# Optimizing Sustainability in Malting Barley: A Practical Approach to Nitrogen Management for Enhanced Environmental, Agronomic, and Economic Benefits

Petros Vahamidis [1,*], Angeliki Stefopoulou [2] and Vassilis Kotoulas [3]

1 Directorate of Plant Production Protection, Hellenic Ministry of Rural Development and Food, 150 Syngrou Avenue, 176 71 Kallithea, Greece
2 Financial Audit Committee, General Accounting Office, Hellenic Ministry of Economy and Finance, 57 Panepistimiou Avenue, 105 64 Athens, Greece; anstef_1@yahoo.gr or a.stefopoulou@edel.gr
3 Athenian Brewery S.A, 102 Kifissos Avenue, 102 10 Athens, Greece; vassilis_kotoulas@heineken.com
* Correspondence: pvachamidis@minagric.gr

**Abstract:** Nitrogen (N) fertilisers used in barley production serve as the primary contributors to total greenhouse gas (GHG) emissions. Consequently, to lower the carbon footprint (CF) and GHG emissions, it is imperative to either reduce N fertiliser rates or enhance grain yield and improve nitrogen use efficiency (NUE). To address this challenge, we combined two strategies related to N: (1) a 34% reduction in the total N rate compared to the control (total N rate 108–110 kg N ha$^{-1}$), and (2) testing two types of N fertilisers for topdressing against the control (common sulfur urea). These types included (a) a mixture comprising controlled-release fertiliser (CRF) combined with ammonium sulfate nitrate fertiliser in a 40:60 ratio (CRF + Nitro) and (b) ammonium sulfate nitrate (Nitro). Experiments were conducted in two distinct areas of Greece specialising in cereal production, aiming to unveil the effects of these strategies on all sustainability aspects of malting barley production. The results showed that although a 34% reduction in N rate did not result in yield penalties or a decrease in grain size, it did have a negative impact on grain protein content (GPC). CRF + Nitro not only reduced CF by approximately 30% compared to the control but also increased N agronomic efficiency by 51.5% and net profit by 7.1%. Additionally, it was demonstrated that the maximum achievable reduction in total GHG emissions and CF, by excluding N fertilisation from the crop system, ranged from 68.5% to 74.3% for GHG emissions and 53.8% to 67.1% for CF.

**Keywords:** malting barley; carbon footprint; nitrogen fertilisation; grain yield; economic analysis; controlled release fertilisers (CRFs); nitrogen use efficiency; grain protein content

## 1. Introduction

Nitrogen (N) plays a crucial role in plant growth and serves as a key determinant of crop yields worldwide. The need for N inputs to sustain yields results in the application of around 137 trillion grammes (Tg) of N per year, of which almost half is contributed by mineral nitrogen fertilisers, and a total nitrogen output of 148 Tg of N per year, of which 55% is uptake by harvested crops and crop residues [1]. Furthermore, it is widely recognised that the recovery of applied N by crops in the year of application is typically low, ranging between 26% and 68% [2–5]. Additionally, there is a general consensus that effective nitrogen management is pivotal for achieving low emissions in cereal production [6]. Indeed, synthetic nitrogen fertilisers, employed in the cultivation of cereal crops, account for the largest percentage (approximately 27–70%) of total GHG emissions [7–14]. Specifically, the application of nitrogen fertilisers to soils stimulates nitrous oxide (N$_2$O) emissions primarily through microbial processes such as nitrification and denitrification [15,16]. N$_2$O is a major GHG with a global warming potential 265 times that of carbon dioxide (CO$_2$)

over a 100-year timescale [17]. Existing evidence indicates a general trend of exponentially increasing $N_2O$ emissions as N inputs increase to exceed crop needs [18].

It has been suggested that, in order to lower the carbon footprint and GHG emissions, it is essential to either reduce N fertiliser rates [14] or enhance grain yield and improve nitrogen use efficiency (NUE) [9]. Regarding the first approach, there are several barriers that we need to overcome. Barley is very sensitive to N availability, so a deficiency of N limits grain yields not only under favourable conditions but also in low-yielding environments [19–21]. Furthermore, underfertilization may have environmental consequences, leading to a decline in soil organic nitrogen and associated soil organic carbon [22] or having a negative impact on crop water use efficiency [21,23]. In malting barley, aside from achieving high yields, specific quality criteria must be met regarding grain protein content (GPC) and grain size to optimise industrial processes [24–26]. In our previous experiments across Greece, we demonstrated that maintaining the total nitrogen supply below 100 kg N ha$^{-1}$ led to a decline in GPC under high-yielding conditions [25]. While the pursuit of identifying new nitrogen fertilisation strategies that achieve both high agronomic and environmental performance is undeniably challenging, there is evidence suggesting that low-N strategies are possible for malting barley [24].

In accordance with the "4R" nutrient stewardship framework, effective N management entails selecting the appropriate fertiliser source, applying the correct rate at the optimal time, and placing it in the right place [27]. In this context, a method to reduce GHG emissions from agricultural fields and enhance nitrogen use efficiency (NUE) involves the utilisation of controlled-release fertilisers (CRFs). These products are specially designed to purposefully release the active nutrients in a controlled manner to coincide with and match the specific nutrient demand during plant growth [28,29]. CRF granules are commonly coated with inorganic, mineral-based coatings such as sulfur and organic coatings using polymers, thermoplastics, or resins [28,30]. It is important to note that the release of nutrients from CRFs commonly depends on the temperature, the soil moisture, and the composition and thickness of the coating materials [31].

The application of CRFs has resulted in increased crop yields, enhanced NUE, and reduced $N_2O$ emissions [32–37]. Despite these benefits, the use of CRFs comes with certain drawbacks. Primarily, the increased production costs of CRFs, in comparison to conventional fertilisers, stand out as a significant limiting factor for their widespread adoption [28]. Additionally, current CRFs exhibit susceptibility to fluctuations in temperature, ambient moisture, soil bioactivity, and the wetting and drying cycles of the soil. Consequently, any alterations in these conditions can lead to an unpredictable release rate of the fertilisers, adversely impacting their efficiency [38–40]. Therefore, to meet crop N requirements over the entire growth period while reducing fertiliser costs in comparison to 100% CRF, recent research has focused on blending controlled-release urea with common urea. This strategy has been found to enhance NUE, reduce fertiliser costs, and improve the environmental performance of various crops such as wheat [14,41,42], potato [43], rice [44], maize [45], and tomato [46].

Building on the favourable outcomes of this approach, we hypothesise that further optimisation can be achieved by replacing common urea with a nitrate-containing fertiliser. This hypothesis is grounded in the observation that fertilisers containing nitrates, such as ammonium nitrate, demonstrate a more substantial positive impact on grain yield in cereals [47,48] and result in lower GHG emissions when compared to common urea fertilisers [49–51]. Therefore, the objective of this study was to unveil the combined effects of a 34% reduction in total nitrogen rate and various nitrogen fertilisation approaches for topdressing (i.e., common urea vs. mix of CRF with ammonium sulphate nitrate vs. ammonium sulfate nitrate) on all sustainability aspects (i.e., GY, GPC, grain size, environmental impact, and economic viability) of malting barley production.

## 2. Materials and Methods

### 2.1. Site Description and Crop Management

The experiments were carried out during the 2022–2023 growing season in two contrasting areas of Greece specialising in cereal production, namely in Livadia (38°26′47.70″ N, 22°54′47.96″ E) and Larissa (39°30′4.14″ N, 22°18′39.18″ E) in commercial fields. Soil properties and crop management in each experiment are presented in Tables S1 and S2. Meteorological data (precipitation, maximum, minimum, and average temperatures) were recorded daily in the experimental areas and are provided in Figure 1. Sowing was carried out during the optimal time period (November-December) at a seeding rate of approximately 350 seeds m$^{-2}$, with a row distance of 12 cm. The two-rowed spring barley (*Hordeum vulgare* L.) Fortuna (Ackermann Saatzucht GmbH & Co. KG, Irlbach, Germany), a widely cultivated malting cultivar in Greece and Europe, was used in both experimental sites.

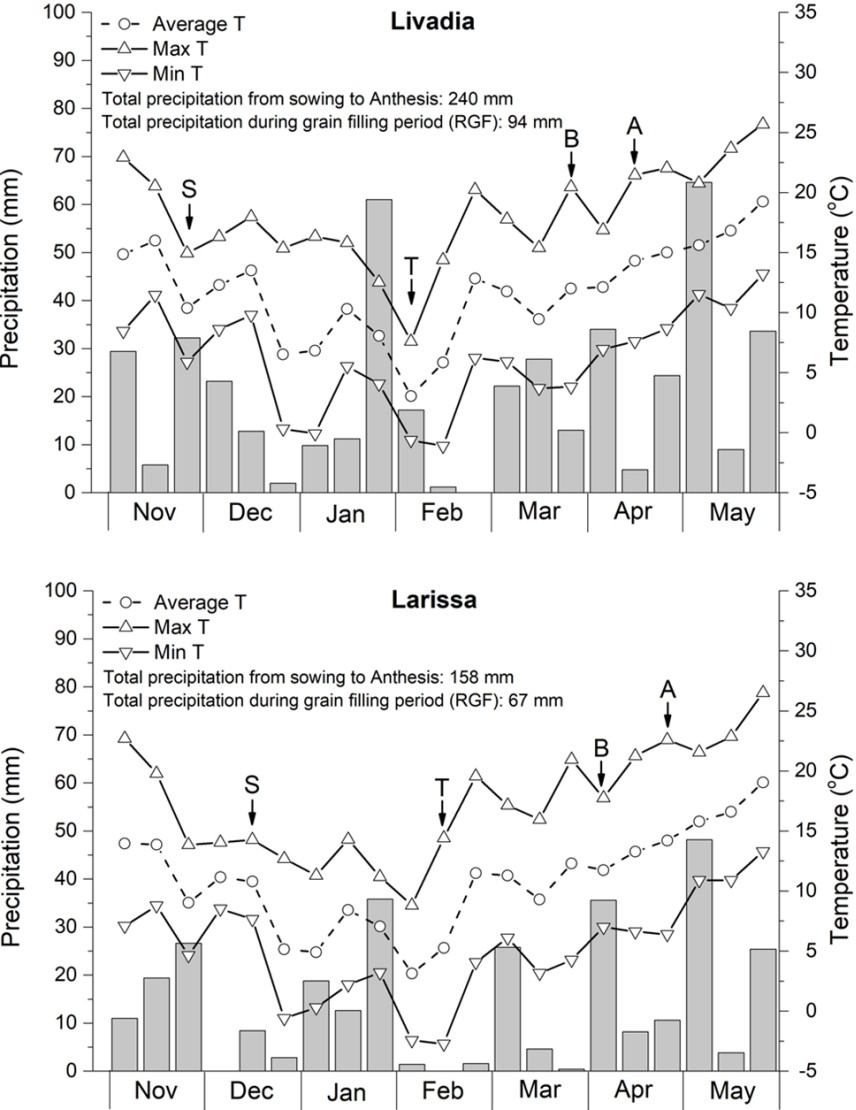

**Figure 1.** Precipitation and air temperature (average, Tmin, and Tmax) profiles for the Larissa and Livadia experiments. Arrows denote key phenological stages: S = sowing; T = tillering; B = booting; A = anthesis. RGF represents the cumulative rainfall during the period spanning 6 days pre-anthesis to 20 days post-anthesis, identified as the primary environmental factor influencing plump grain (or retention fraction) determination [52].

### 2.2. Treatments and Experimental Design

The experiments were arranged in a randomised complete block design (RCB), with four replicates in each treatment. The plot size for each treatment was 0.09 ha in Livadia and 0.18 ha in Larissa. The specific objective of these experiments was twofold: (1) to investigate the impact of total nitrogen reduction from both basal and topdressing applications, and (2) to evaluate the effects of different types of nitrogen fertilisers used for topdressing. Consequently, this study does not specifically focus on the fertilisers used for basal fertilisation. However, it was deemed crucial to maintain consistent amounts of phosphorus and potassium in all tested treatments to ensure that nitrogen remains the main source of variation. The treatments employed were as follows: (N0) no nitrogen fertiliser; (Control) following the farmer's conventional fertilisation practice, applying a total of 108–110 N kg ha$^{-1}$ in a split application and using common sulfur urea for topdressing; (CRF + Nitro) featuring a 34% reduction in total nitrogen using a mix of controlled release fertiliser with ammonium sulfate nitrate fertiliser (in a 40:60 ratio) as topdressing; and (Nitro) with a 34% reduction in total nitrogen using ammonium sulfate nitrate fertiliser as topdressing. The decision to opt for a 34% reduction in total nitrogen is grounded in our preliminary experiments conducted at the same experimental sites in the preceding growing season (2021–2022). These experiments revealed that a reduction in nitrogen dosage by approximately 40% (when comparing the control to the low-carbon practices tested) resulted in a significant decrease in carbon footprint (CF), ranging from 16.7% to 26.3%. However, this reduction in nitrogen dosage was accompanied by a decline in grain yield, ranging from −8.1% to −15% compared to the control. Consequently, the primary target of the current experiments was to sustain the positive outcomes in carbon footprint while simultaneously improving grain yield.

The ratio of basal N (applied during sowing) to topdressing N (applied at the tillering stage) was 15:85 for CRF + Nitro and Nitro in Livadia, 25:75 for CRF + Nitro and Nitro in Larissa, and 19:81 and 33:67 for the control in Livadia and Larissa, respectively. The CRF fertiliser used was Agromaster® (ICL), characterised by a polyurethane coating with a nitrogen content of 40% (4.0% as ammonium-N and 36.0% as urea-N), a sulfur content of 12%, and a controlled release period of 60–90 days. In the CRF + Nitro treatment, CRF and ammonium sulfate nitrate were applied in two separate passes—one for the CRF and another for the ammonium sulfate nitrate—to ensure uniform spreading. Details of N fertiliser strategies are provided in Table 1.

**Table 1.** Description of N treatments in this study.

| Location | Treatment | Basal Fertilisation | | N Topdressing * | | Total N Rate (kg N ha$^{-1}$) |
|---|---|---|---|---|---|---|
| | | Fertiliser | N Rate (kg ha$^{-1}$) | Fertiliser | N Rate (kg ha$^{-1}$) | |
| Livadia | Control | 11-15-15 | 20 | Sulfur urea (40-0-0) | 88 | 108 |
| | CRF + Nitro | 9-22-22 | 11 | CRF      ASN | 60 | 71 |
| | Nitro | 9-22-22 | 11 | ASN | 60 | 71 |
| | N0 | 0-46-0      0-0-60 | 0 | - | - | 0 |
| Larissa | Control | 20-10-0 | 36 | Sulfur urea (40-0-0) | 74 | 110 |
| | CRF + Nitro | 16-20-0 | 18 | CRF      ASN | 55 | 73 |
| | Nitro | 16-20-0 | 18 | ASN | 55 | 73 |
| | N0 | 0-46-0 | 0 | - | - | 0 |

* N topdressing application took place at GS 22 and GS 24 in Larissa and Livadia, respectively. CRF: controlled-release fertiliser containing urea (40-0-0); ASN: ammonium sulfate nitrate (26-0-0). CRF and ASN in CRF + Nitro treatments were applied in two separate passes—one for the CRF and another for the ammonium sulfate nitrate—to ensure even spreading.

### 2.3. Measurements

Phenology was monitored weekly using the scale of Zadoks et al. [53], following the average phenology in each plot (when 50% of the plants reached the developmental stage).

### 2.3.1. Ground Cover

The green canopy cover in both experiments was assessed using the mobile phone app Canopeo [54]. Canopeo is an automatic colour threshold image analysis tool developed in the Matlab programming language using colour values in the red–green–blue (RGB) system. Canopeo photos were taken freehand at approximately 1.3 m above ground level, under natural field illumination conditions [55] from 10 positions in each plot, using a smartphone camera (Samsung[TM] (Suwon, Republic of Korea): Galaxy S22, 50.0 megapixels, 1848 × 4000 pixels).

### 2.3.2. Yield and Yield Components

Crop yields (adjusted to 11% of moisture content and expressed on a t/ha basis) were determined by harvesting the entire plot with a commercial combine harvester at maturity (11.6 and 10.7% of grain moisture content in Livadia and Larissa, respectively). For the determination of yield components, three quadrats of 1 m$^2$ were manually sampled from each plot following a Z pattern prior to harvesting. Spikes/m$^2$ were determined from the total samples in each plot. The grain number per spike was determined from 30 randomly selected spikes per plot. For the thousand kernel weight (TKW) determination, two samples per plot, consisting of 200 entire clean grains, were counted, and their weight was expressed on a g/1000 grain basis. When the two samples differed by more than 10%, a third sample was taken. All grains, irrespective of their size, were included to determine TKW.

### 2.3.3. Nitrogen Agronomic Efficiency (NAE)

N-agronomic efficiency (NAE) was calculated according to Duan et al. [56]:

$$NAE = (Y_N - Y_0)/A_N, \tag{1}$$

where $Y_N$ (kg grain ha$^{-1}$) is the grain yield from treatments with N fertiliser, $Y_0$ (kg grain ha$^{-1}$) is the grain yield from treatments without N fertiliser, and $A_N$ (kg N ha$^{-1}$) is the amount of fertiliser applied.

### 2.3.4. Grain Size

Grain size was determined through size fractionation using a screening machine equipped with three slotted sieves of different widths (2.8, 2.5, and 2.2 mm), following the Analytica EBC "Sieving Test for Barley" method [57]. A 100 g sample of grain was placed on the top sieve (2.8 mm) and shaken for 5 min. Each grain sample was then sorted into four grain size fractions: >2.8 mm (fraction 1), 2.8–2.5 mm (fraction 2), 2.5–2.2 mm (fraction 3), and <2.2 mm (fraction 4), using a Sortimat machine (Pfeuffer GmbH, Kitzingen, Germany).

### 2.3.5. Grain Protein Content (GPC)

Nitrogen content was determined using the Kjeldhal method, and protein content was calculated by multiplying the N content by a factor of 6.25.

### 2.4. Carbon Footprint (CF) and Greenhouse Gas Emissions (GHG) Calculation

CF was estimated using the Cool Farm Tool (CFT) v2.5 (https://coolfarmtool.org/ (accessed on 20 September 2023)) [58]. The CFT serves as a GHG calculator designed for product-level calculations, specifically for determining emissions associated with individual products produced on-farm, such as potatoes [59], wheat [60,61], rice [62,63], maize [64], and cacao [65]. Total carbon emissions for crops encompass various factors [66], including: (1) methane and nitrous oxide emissions from crop residues; (2) emissions from fertiliser production (Fertilisers Europe CFP calculator); (3) emissions from soil due to the application of N-fertilisers (direct and indirect emissions modelled based on Tier 1 IPCC); (4) emissions from pesticide use (World Food Lifecycle Database); (5) emissions from machinery and energy use; (6) emissions associated with seed production; (7) emissions from waste water;

(8) emissions due to changes in carbon stocks; (9) emissions due to N mineralized in mineral soils as a result of loss of soil carbon; and (10) transport emissions. In comparison to other carbon accounting tools, CFT was recently identified as the highest-ranking tool currently available in the public domain [67].

The total greenhouse gas emissions (GHG) calculation in CFT was based on the following equation [66]:

$$L_{croptotal} = L_{residue} + L_{fert.prod.} + L_{fert.} + L_{pest.} + L_{fuel\&energy} + L_{trans} \tag{2}$$

where $L_{croptotal}$ total crop GHG emissions [kg $CO_2$ eq], $L_{residue}$ nitrous oxide emissions from crop residues [kg $CO_2$ eq], $L_{fert.prod.}$ emissions from fertiliser production [kg $CO_2$ eq], $L_{fert.}$ emissions from soil due to application of N-fertilisers [kg $CO_2$ eq], $L_{pest.}$ emissions from pesticide use [kg $CO_2$ eq], $L_{fuel\&energy}$ emissions from machinery and energy use [kg $CO_2$ eq], and $L_{trans}$ transport emissions [kg $CO_2$ eq].

The proportion of residues that are left on the field after harvest contains nitrogen, which contributes to $N_2O$ emissions ($L_{residue}$; Equation (5)). The amount of added nitrogen was calculated from the sum of above and below ground content (Equation (3)), and then the relevant emission factors were applied for direct and indirect emissions (Equation (4)) [66].

$$N_{residue} = R_{above} \times N_{AG(T)} + R_{below} \times N_{BG(T)} \tag{3}$$

$$L_{residue,N_2O-soil} = \left( (N_{residue} \times EF_1) + \left( N_{residue} \times EF_5 \times Frac_{LEACH-(H)} \right) \right) \times \frac{44}{28} \tag{4}$$

$$L_{residue} = L_{residue,N_2O\text{-}soil} \times GWP_{N_2O} \tag{5}$$

where $N_{residue}$ nitrogen content in residues [Kg], $R_{above}$ amount of above-ground residues that were left on the field after harvest [Kg], $N_{AG(T)}$ fraction of N in above-ground residues (we used the default value 0.007; IPCC, 2019, Geneva, Switzerland), $R_{below}$ amount of below-ground residues (we used the default ratio of below-ground root biomass to above-ground shoot biomass that is 0.22; IPCC, 2019) [Kg], $N_{BG(T)}$ fraction of N in below-ground residues (we used the default value 0.014; IPCC, 2019), $L_{residue,N_2O\text{-}soil}$ $N_2O$ emissions from N in crop residues returned to the soil [kg ($N_2O$)], $EF_1$ direct $N_2O$ emission factor (0.005, in dry climates; IPCC, 2019), $EF_2$ indirect $N_2O$ emission factor (NA, in dry climates; IPCC, 2019), $Frac_{LEACH\text{-}(H)}$ fraction of all N added to/mineralized in managed soils in regions where leaching/run-off occurs that is lost through leaching and run-off (0, in dry climates; IPCC, 2019), 44/28 Conversion of $N_2O$–N emissions to $N_2O$ emissions and $GWP_{N_2O}$ global warming potential for $N_2O$ (273; [66]) [kg $CO_2$ eq].

Total GHG emissions from fertiliser production were calculated from Equation (6) [66]:

$$L_{fert.prod.} = \sum_i R_i \times EF_{fert}(i) \tag{6}$$

where $L_{fert.prod.}$ emissions from fertiliser production [kg $CO_2$ eq], $R_i$ application rate of fertiliser type $i$ [Kg], and $EF_{fert}(i)$ emissions factor of fertiliser type $i$ [kg ($CO_2$ eq)kg$^{-1}$]. For urea and ammonium sulphate nitrate, the production emission factor was 0.878009263 and 0.797155511, respectively (Fertilisers Europe Calculator). Regarding the fertilisers 11-15-15, 9-22-22, 20-10-0, and 16-20-0, the CFT also includes a method for estimating fertiliser production emissions based on the ratio of ingredients and the production location (Equation (7)) [66]:

$$EF_{custom\,fert.} = EF_i + \sum_j P_j \times EF_{ingredient}(j) \tag{7}$$

where $EF_{custom\,fert.}$ emissions from fertiliser production [kg $CO_2$ eq], $EF_i$ overall production emissions factor for composing your own fertiliser [kg ($CO_2$ eq)kg$^{-1}$], $P_j$ proportion of ingredient $j$ and $EF_{ingredient}(j)$ emission factor for production of ingredient $j$ in Europe (0.413667 for $K_2O$; 2.78882141 for Ammonia; 3.64256191 for Nitric acid; 1.90871579 for Urea; 0.12460127 for $P_2O_5$).

Direct fertiliser-induced emissions were modelled based on a Tier 1 IPCC (2019) approach (Equation (8)) [66]:

$$L_{N_2O \; direct} = \sum_n (A_n \times EF_n) \frac{44}{28} \tag{8}$$

where $L_{N_2O \; direct}$ direct $N_2O$ emissions [kg ($N_2O$)], $A_n$ applied N via synthetic fertiliser [kg (N)], $EF_n$ emission factor for direct emissions from nitrogen application (we used the default value 0.005 for dry climates; IPCC, 2019), and 44/28 conversion of $N_2O$–N emissions to $N_2O$ emissions.

The direct $N_2O$ emissions for the CRF ($L_{direct \; CRF}$) were calculated from Equation (9) [66]:

$$L_{direct \; CRF} = L_{N_2O \; direct} \times 0.63 \tag{9}$$

where $L_{N_2O \; direct}$ direct $N_2O$ emissions from Equation (8) and 0.63 is a default factor derived from Akiyama et al. [32].

Indirect fertiliser-induced emissions (i.e., via leaching and volatilization) were calculated from Equation (10) [66]:

$$L_{N_2O \; indirect} = L_{N_2O \; leach.} + L_{vol.} \tag{10}$$

Leaching and volatilization emissions were modelled based on Tier 1 IPCC (2019) from Equations (11) and (12), respectively [66]:

$$L_{N_2O \; leach.} = A_{total} \times Frac_{LEACH-(H)} \times EF_5 \times \frac{44}{28} \tag{11}$$

$$L_{vol.} = \sum_n (A_n \times F_{vol,\,n} \times EF_4) \frac{44}{28} \tag{12}$$

where $L_{N_2O \; leach.}$ emissions via leaching [kg ($N_2O$)], $A_{total}$ total applied N via synthetic fertiliser [kg (N)], $Frac_{LEACH\text{-}(H)}$ Leaching factor set at 0, $EF_5$ Emission factor for leaching/runoff set at 0.011, $L_{vol.}$ emissions via volatilization, $A_n$ total applied nitrogen for fertiliser $n$ [kg (N)], $F_{vol,n}$ volatilization factor for fertiliser $n$ (urea = 0.15; ammonium sulphate nitrate = 0.0625; NPK fertiliser = 0.11; P fertiliser = 0; K fertiliser = 0), $EF_4$ emission factor for volatilization set at 0.005 and 44/28 conversion of $N_2O$–N emissions to $N_2O$ emissions.

Emissions from urea ($L_{urea}$) were calculated from Equation (13) [66]:

$$L_{urea} = (A_{urea} \times EF_{urea}) \frac{44}{12} \tag{13}$$

where $A_{urea}$ total applied urea [kg (urea)], $EF_{urea}$ emission factor for urea set at 0.20 kg $CO_2$-C t$^{-1}$ and 44/12 conversion of $CO_2$-C emissions to $CO_2$ emissions.

Emissions via pesticides ($L_{pest}$) were calculated according to Equation (14) [66]:

$$L_{pest} = A \times R_{app.} \times AI_{frac} \times EF_{pest} \tag{14}$$

where, $A$ area [ha], $R_{app.}$ application dose [kg ha$^{-1}$], $AI_{frac}$ fraction of active ingredient, and $EF_{pest}$ emission factor per type of pesticide [kg $CO_2$ eq kg$^{-1}$] (fungicide = 8.2977 kg $CO_2$ eq; herbicide = 8.7346 kg $CO_2$ eq).

Total emissions through fuel consumption ($L_{fuel}$) for farm operations (e.g., tillage, sowing, pesticide spraying, spreading fertiliser, harvesting) were estimated from Equation (15) [66]:

$$L_{fuel} = \sum_i C_{fuel} \times EF_{fuel} \tag{15}$$

where $C_{fuel}$ consumption of fuel $i$ [litre] and $EF_{fuel}$ emission factor of fuel are set at 3.33427 kg $CO_2$ eq/litre.

The total emissions resulting from the transportation ($L_{transport}$) of farm inputs such as fertilisers and pesticides to the farm gate were calculated using Equation (16) [66]:

$$L_{transport} = EF_{VEH} \times D \times M \tag{16}$$

where $EF_{VEH}$ emission factor for transport specific to vehicle type [kg $CO_2$ eq $t^{-1}$ $km^{-1}$] is set at 0.10749 kg $CO_2$ eq $t^{-1}$ $km^{-1}$, $D$ distance to farm gate [km] and $M$ transported mass [t].

The total greenhouse gas (GHG) emissions were computed by summing the accumulated $CO_2$ equivalents from all the sources mentioned above (Equation (2)). Subsequently, the carbon footprint (expressed as kg $CO_2$ equivalent per kg of grain) for malting barley was determined as follows [13,68]:

$$CF = \frac{total\ GHG\ emissions}{grain\ yield} \tag{17}$$

### 2.4.1. System Boundary

The system boundary was set from 'cradle to farm gate' (Figure 2), considering all the production processes involved, from raw material extraction (i.e., the cradle) to the point where the final product reached the farm gate (harvested barley). The following activities were included: (1) synthetic nitrogen (N), phosphorus (P), and potassium (K) fertilisers used in the processes of manufacturing, transportation (i.e., from the industry facilities in Greece to the farm gate), and application; (2) crop residue decomposition; (3) N loss via volatilization and leaching; (4) production, transportation (i.e., from the industry facilities in Greece to the farm gate), and application of herbicides and fungicides; and (5) fuel consumption in different field activities, from land preparation until harvesting and bailing of barley straw. The production of agricultural machines and crop seeds was not considered due to a lack of data and also because of the similarity among fertiliser treatments, which could not affect the main goals of the present study [69].

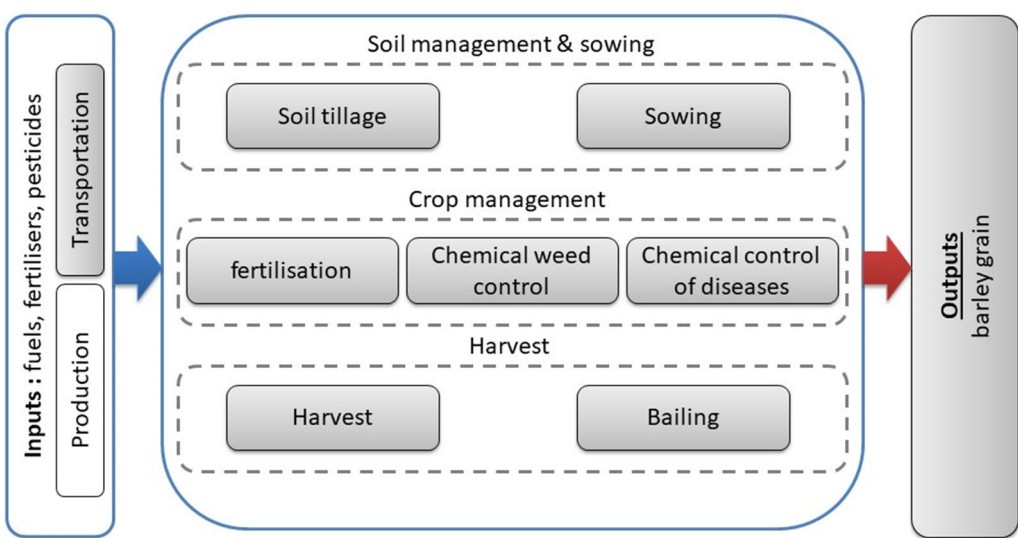

**Figure 2.** System boundary of barley production. The production of agricultural machines and crop seeds was not included in CF calculation. Processes modelled with primary data are represented with a grey background, while those relying on secondary data are depicted with a white background.

The changes in the soil organic matter content (SOC) due to the imposed treatments were not included in the analysis because it is technically difficult to quantify SOC changes within a year due to the spatial variability and detection limits of analytical methods [6]. Thus, it is suggested that long-term field experiments, lasting many years or decades,

are needed for detecting any change in SOC sequestration within various land and soil management options [70].

Primary data from the two experimental farms (Tables 1, S1 and S2) were paired with secondary data taken from the CFT, including inventories for the production of fertilisers and pesticides, emission factors related to fuel consumption during farm operations (e.g., tillage, sowing, spraying, etc.), and emission factors associated with the transportation of inputs (such as fertilisers and pesticides), specific to the type of vehicle used.

2.4.2. Functional Units

Two functional units were used for C footprint calculation, namely carbon footprint per unit area expressed as kg $CO_2$-equivalents per hectare (kg $CO_2$ eq/ha) and carbon footprint per unit weight expressed as kg $CO_2$-equivalents per kg of grain (kg $CO_2$ eq/kg grain) (11% moisture content) at the farm gate.

*2.5. Economic Analysis*

The net profit (EUR ha$^{-1}$) of grain production was estimated using the following equation [13]:

$$Net\ profit = Yield \times P - \sum_{i=1}^{m} C_i \qquad (18)$$

where P (EUR kg$^{-1}$) is the local in-season purchasing price of barley (0.24 EUR kg$^{-1}$ grain) and $C_i$ is the cost (VAT-exclusive prices) of various farming inputs (seeds, fuel for farming operations, fertilisers, herbicides, fungicides, and harvesting). The input cost was calculated according to the current prices.

*2.6. Statistical Analysis*

The effects of location and fertiliser treatments on the tested variables were assessed through the analysis of combined experiments [71] using the Statgraphics Centurion ver. XVI software package (Statpoint Technologies, Inc., Warrenton, VA, USA). Before conducting ANOVA, standardised residuals were visually examined with qq-plots as well as with the Shapiro–Wilk test using SPSS (IBM SPSS Statistics for Windows, Version 22.0, IBM Corp. Armonk, New York, NY, USA). Location was treated as a random factor, while fertiliser treatments were considered fixed effects. Concerning the green canopy cover, the data from the two experiments were analysed separately. This was done due to the differing developmental stages of barley between the two experiments at the time of the observations. Prior to ANOVA, the arcsine transformation was applied to traits measured as a percentage, such as ground cover and grain protein content, and the back-transformed means for these traits are reported. Significant differences between treatment means were compared using the protected least significant difference (LSD) procedure at $p < 0.05$.

**3. Results**

*3.1. Green Canopy Cover*

The analysis of variance revealed that the application of nitrogen in basic fertilization significantly increased green canopy cover only in Larissa ($p < 0.01$) (Figure 3). It is important to highlight that the treatments labelled "CRF + Nitro" and "Nitro" were identical up to the initial observation (i.e., early tillering), diverging only after the application of spring topdress nitrogen. Therefore, the slight and non-significant differences among them during this particular phase reflected the inherent variability in the crop.

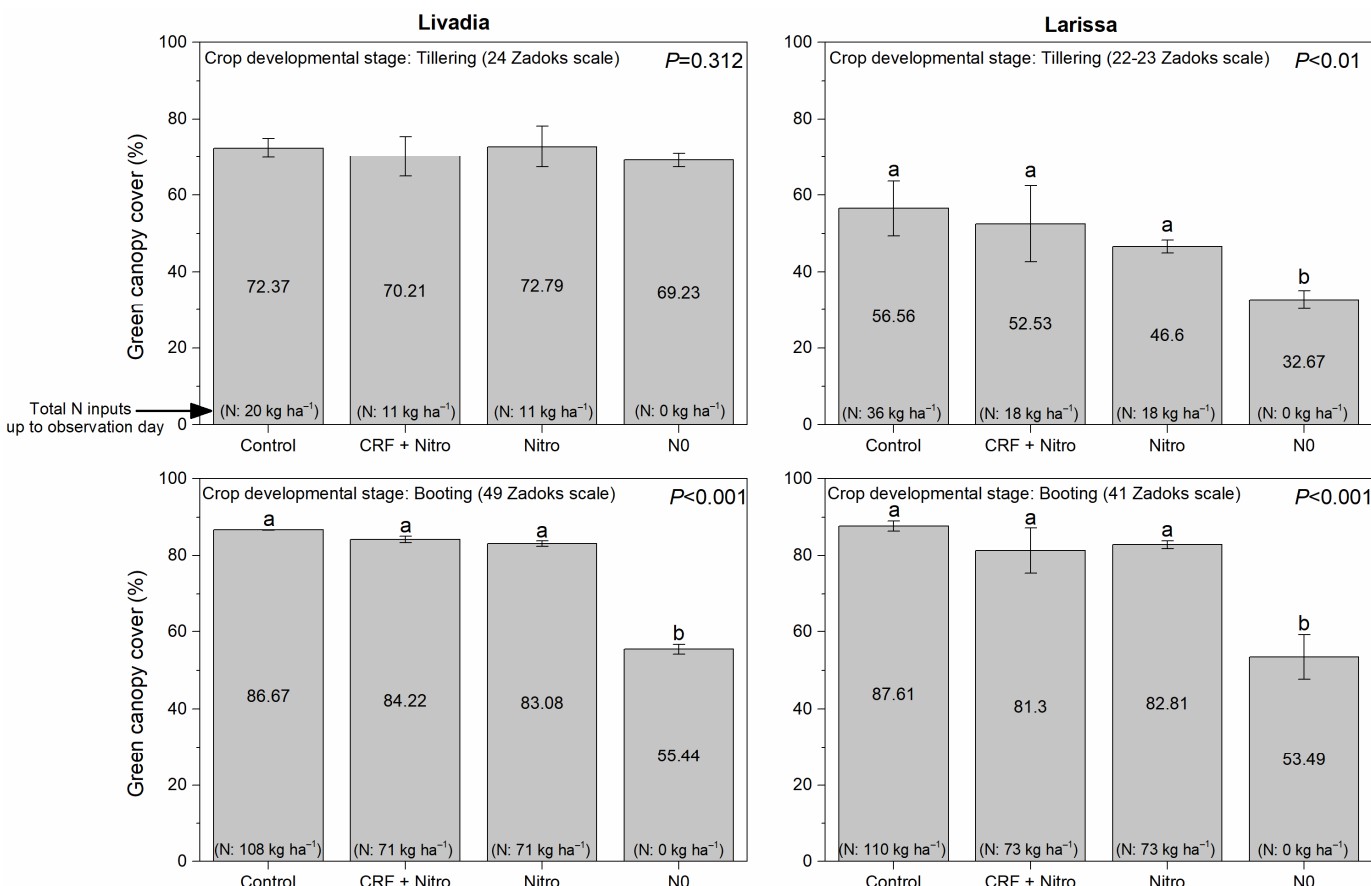

**Figure 3.** Effect of the N fertilization strategies on green canopy cover in Livadia and Larissa. The treatments labelled "CRF + Nitro" and "Nitro" were identical during the first observation, diverging only following the application of spring topdress nitrogen. Error bars represent the standard error of the mean (*n* = 4). Different letters indicate statistical significance differences within the same crop developmental stage and the same location according to L.S.D. test (*p* < 0.05).

In the second observation phase, which corresponds to the booting stage (i.e., early and late booting in Larissa and Livadia, respectively), it was found that the tested nitrogen practices exerted a significant effect on crop green canopy cover in both experiments (Livadia: *p* < 0.001; Larissa: *p* < 0.001). Overall, there were no significant differences in terms of green canopy cover between the tested low-carbon practices ("CRF + Nitro" and "Nitro") and the control plots during both observation periods, namely tillering and booting (Figure 3).

### 3.2. Yields and Nitrogen Agronomic Efficiency (NAE)

Grain yield (GY) was significantly affected by treatment (*p* < 0.05), location (*p* < 0.001), and the interaction location × treatment (*p* < 0.001) and ranged between 4.71 t ha$^{-1}$ and 7.63 t ha$^{-1}$ (Table 2). In contrast to the control, the N dosage in the tested low-carbon farming practices ("CRF + Nitro" and "Nitro") exhibited a reduction of approximately 34%. However, none of the examined low-carbon practices showed statistically significant differences from the control in terms of GY. This can be attributed to the lack of significant differences between the control and the low-carbon practices concerning grains spike$^{-1}$ and spikes m$^{-2}$. Biomass followed the same trend. Furthermore, location had a significant effect on all yield traits except thousand kernel weight (TKW).

**Table 2.** Grain yield (GY), yield components, and plant height are affected by the N fertilisation strategies. Analysis of variance (ANOVA) is also shown.

| Location | Treatment | GY (t ha$^{-1}$) | Grains Spike$^{-1}$ | Spikes m$^{-2}$ | Biomass (t ha$^{-1}$) | TKW (g) | Plant Height (cm) |
|---|---|---|---|---|---|---|---|
| Livadia | Control | 7.28 ab | 23.7 a | 669 a | 13.92 a | 45.9 a | 76.4 a |
| | CRF + Nitro | 7.22 a | 24.2 a | 615 a | 13.35 a | 48.5 b | 72.7 a |
| | Nitro | 7.63 b | 24.4 a | 650 a | 13.43 a | 48.1 b | 75.2 a |
| | N0 | 4.99 c | 20.2 b | 508 b | 8.92 b | 48.6 b | 58.4 b |
| Larissa | Control | 6.21 a | 24.9 a | 514 a | 12.20 a | 48.5 a | 72.5 a |
| | CRF + Nitro | 6.25 a | 25.0 a | 510 a | 12.07 a | 49.0 ab | 75.0 a |
| | Nitro | 5.94 a | 26.2 a | 455 a | 12.02 a | 49.9 b | 77.7 a |
| | N0 | 4.71 b | 26.2 a | 369 b | 8.12 b | 48.7 a | 59.2 b |
| ANOVA | | | | | | | |
| Location (L) | | *** | *** | *** | * | ns | ns |
| Treatment (T) | | * | ns | * | ** | ns | ** |
| L × T | | *** | *** | *** | ns | * | ns |

ns: non-significant effect; TKW: thousand kernel weight. Different letters indicate statistical significance differences within the same column and the same location according to L.S.D. test ($p < 0.05$). * F values significant at the $p < 0.05$ probability levels. ** F values significant at the $p < 0.01$ probability levels. *** F values significant at the $p < 0.001$ probability levels.

NAE was significantly affected by treatment ($p < 0.05$), location ($p < 0.01$), and the interaction location × treatment ($p < 0.01$) and ranged between 13.68 kg kg$^{-1}$ N under the control in Larissa and 37.11 kg kg$^{-1}$ N under the Nitro treatment (i.e., fertilisation with ammonium sulphate nitrate) in Livadia (Figure 4). In comparison to the control, the application of CRF + Nitro and Nitro led to substantial increases in NAE of 48% and 76%, respectively, in Livadia. Meanwhile, in Larissa, the CRF + Nitro treatment exhibited the highest NAE, displaying statistically significant differences when compared to both the Control and Nitro.

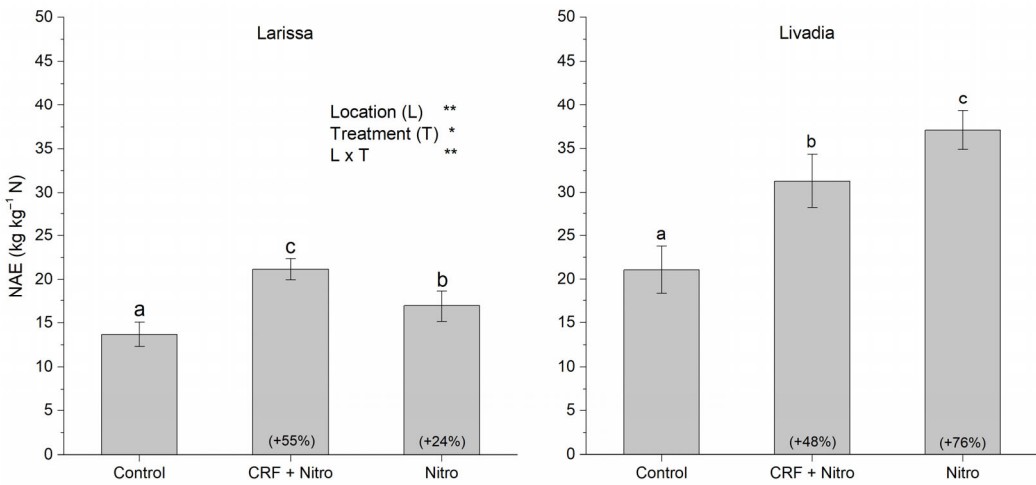

**Figure 4.** Nitrogen agronomic efficiency (NAE) across the different N fertilisation strategies in Livadia and Larissa. Error bars represent the standard error of the mean (*n* = 4). Different letters indicate statistical significance differences within the same location according to L.S.D. test ($p < 0.05$). Percentage change relative to the control is indicated within parentheses. Analysis of variance (ANOVA) is also shown. * F values significant at the $p < 0.05$ probability levels. ** F values significant at the $p < 0.01$ probability levels.

### 3.3. Quality Characters (Grain Size and Grain Protein Content)

The analysis of variance showed that grain protein content (GPC) was significantly affected by treatment ($p < 0.05$) and location ($p < 0.05$). In both experiments, the control

plots exhibited higher GPC compared to the other treatments, with statistically significant differences. In comparison to the control, reducing the nitrogen rate by 34% led to decreases of 12.4% and 11.4% under CRF + Nitro and Nitro, respectively, in Livadia. Conversely, in Larissa, a 34% reduction in the nitrogen rate resulted in decreases of 8.0% and 11.6% under CRF + Nitro and Nitro, respectively. There were no significant differences in GPC among CRF + Nitro, Nitro, and N0.

With the exception of the grain size fraction 2.5–2.8 mm, location and treatment did not exert significant effects on grain size fractions (Table 3). Nevertheless, the location × treatment interaction was significant for the retention fraction ($p < 0.05$), as well as the fractions 2.2–2.5 mm ($p < 0.01$) and <2.2 mm ($p < 0.05$). The control plots exhibited a decreased retention fraction in comparison to the low-carbon practices ("CRF + Nitro" and "Nitro"); however, this difference was not statistically significant when compared to the CRF + Nitro treatment in Larissa. Furthermore, no statistically significant differences were recorded between CRF + Nitro and Nitro regarding the retention fraction.

**Table 3.** Grain protein content (GPC) and sieving test of malting barley as affected by the N fertilisation strategies. Analysis of variance (ANOVA) is also shown.

| Location | Treatment | GPC (%) | Retention (>2.5 mm) | >2.8 mm | 2.5–2.8 mm | 2.2–2.5 mm | <2.2 mm |
|---|---|---|---|---|---|---|---|
| Livadia | Control | 9.37 a | 82.48 a | 55.35 | 27.13 | 12.14 a | 2.42 a |
| | CRF + Nitro | 8.21 b | 88.99 b | 61.61 | 27.38 | 8.44 b | 0.96 b |
| | Nitro | 8.30 b | 88.02 b | 62.33 | 25.69 | 8.11 bc | 1.42 b |
| | N0 | 7.62 b | 89.26 b | 59.39 | 29.87 | 6.80 c | 1.10 b |
| Larissa | Control | 11.20 a | 89.03 a | 70.84 | 18.19 | 7.03 a | 1.59 a |
| | CRF + Nitro | 10.30 b | 90.19 ab | 69.14 | 21.06 | 7.08 a | 1.49 a |
| | Nitro | 9.90 b | 92.37 b | 77.87 | 14.51 | 5.33 b | 1.42 a |
| | N0 | 10.20 b | 89.44 a | 73.29 | 16.16 | 6.80 a | 1.90 a |
| ANOVA | | | | | | | |
| Location (L) | | * | ns | ns | * | ns | ns |
| Treatment (T) | | * | ns | ns | ns | ns | ns |
| L × T | | ns | * | ns | ns | ** | * |

ns: stands for non-significant effect. Different letters indicate statistical significance differences within the same column and the same location according to L.S.D. test ($p < 0.05$). * F values significant at the $p < 0.05$ probability levels. ** F values significant at the $p < 0.01$ probability levels.

*3.4. Life Cycle Emissions*

Carbon footprint (CF), as well as the total GHG emissions, were significantly affected by treatment (CF: $p < 0.05$; GHG: $p < 0.001$), location (CF: $p < 0.05$; GHG: $p < 0.001$), and the interaction location × treatment (CF: $p < 0.001$; GHG: $p < 0.001$) (Figure 5), and varied from 0.06 to 0.19 kg $CO_2$ eq $kg^{-1}$ grain and 301 to 1270 kg $CO_2$ eq $ha^{-1}$, respectively. An approximate 34% reduction in nitrogen dosage (as observed between the control and CRF + Nitro, or control and Nitro) led to a statistically significant decline in total GHG emissions. Specifically, this reduction varied from 29.2% to 30.4% under the CRF + Nitro treatment and from 24.1% to 27.9% under the Nitro treatment. Moreover, the implementation of the tested low-carbon strategies also yielded a notable reduction in CF. This reduction ranged from 27.7% to 31.6% in the case of the CRF + Nitro treatment and from 22.4% to 30.6% regarding the Nitro treatment. The maximum achievable reduction in total GHG emissions and CF by excluding nitrogen fertilisation from the crop system (N0) presented a range of 68.5% to 74.3% for GHG emissions and 53.8% to 67.1% for CF.

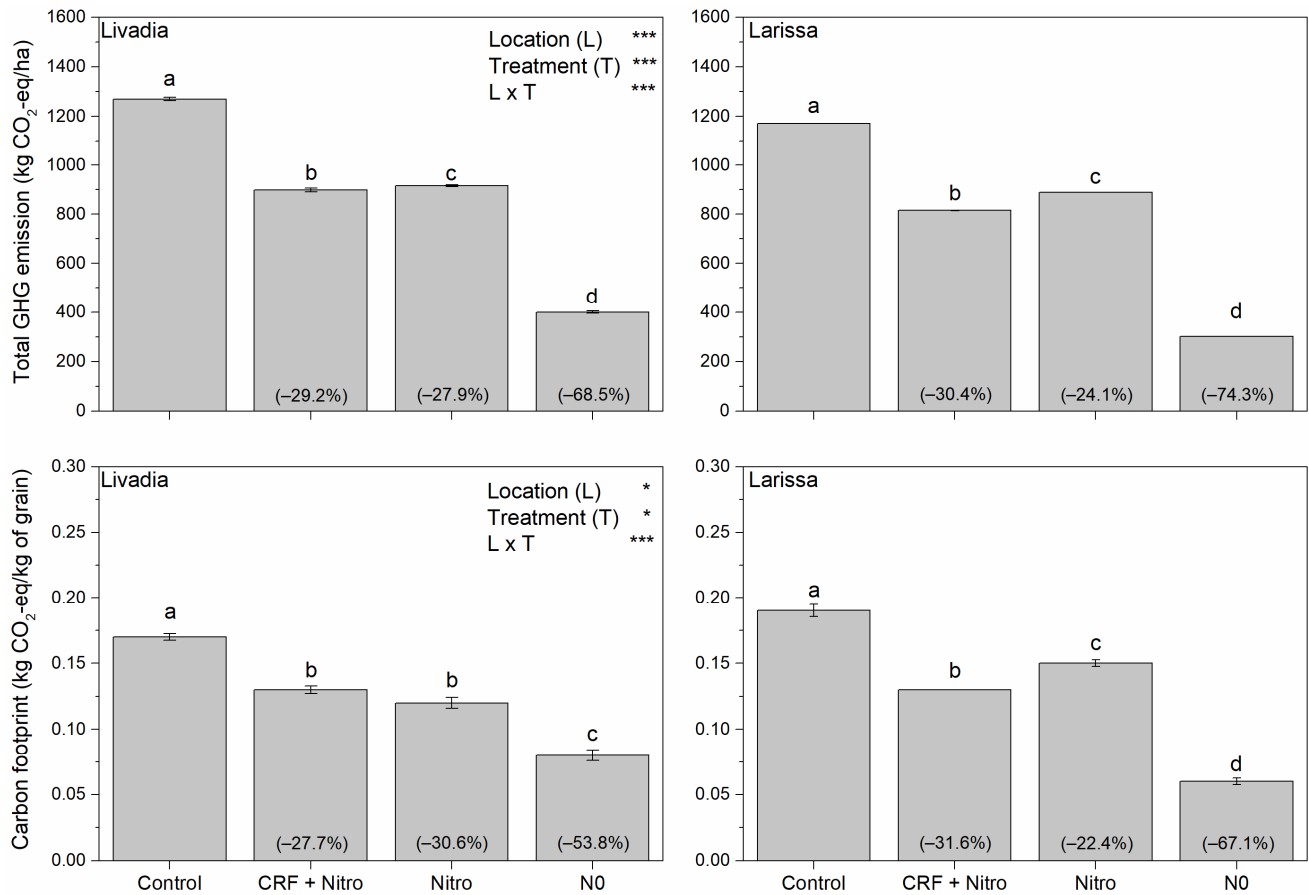

**Figure 5.** Total GHG emissions and carbon footprint across the different N fertilisation strategies in Livadia and Larissa. Error bars represent the standard error of the mean ($n = 4$). Different letters indicate statistical significance differences within the same location according to L.S.D. test ($p < 0.05$). Percentage change relative to the control is indicated within parentheses. Analysis of variance (ANOVA) is also shown. * F values significant at the $p < 0.05$ probability levels. *** F values significant at the $p < 0.001$ probability levels.

The CRF + Nitro and Nitro treatments displayed contrasting responses across the two experiments with respect to CF reduction. In particular, the CRF + Nitro treatment demonstrated superior performance under the conditions of Larissa. However, in Livadia, it was the Nitro treatment that exhibited superior performance over the CRF + Nitro. Even though CRF + Nitro produced lower total GHG emissions in Livadia compared to Nitro (899 kg $CO_2$ eq ha$^{-1}$ vs. 916 kg $CO_2$ eq ha$^{-1}$), this difference did not translate into correspondingly lower CF.

For each treatment, the relative contributions of the different input factors to GHG emissions were consistent across the two experiments (Figures 6 and 7). With the exception of the N0, the highest contribution to total GHG emissions per ha came from soil fertiliser-induced emissions (i.e., production of fertilisers and direct and indirect soil $N_2O$ emissions) and ranged from 57.94% (fertiliser production: 22.89% + fertiliser application: 35.05%) under the CRF + Nitro in Livadia to 75.51% (fertiliser production: 22.48% + fertiliser application: 53.03%) under the control in Larissa. As the total amount of synthetic nitrogen decreased, the contribution of farm operations to total GHG emissions increased. Indeed, with the exception of the control in both experiments, the second largest contribution (22.66–88.27%) to GHG emissions was $CO_2$ from fossil fuel combustion during farm operations (tillage, harvest, etc.). Off-farm transport presented the lowest contribution (≤0.79%) to GHG emissions among all sources, followed by crop protection (Larissa: spraying with

herbicides; Livadia: spraying with herbicides and fungicides) (0.25–1.29%) and crop residue decomposition (1.11–2.46%).

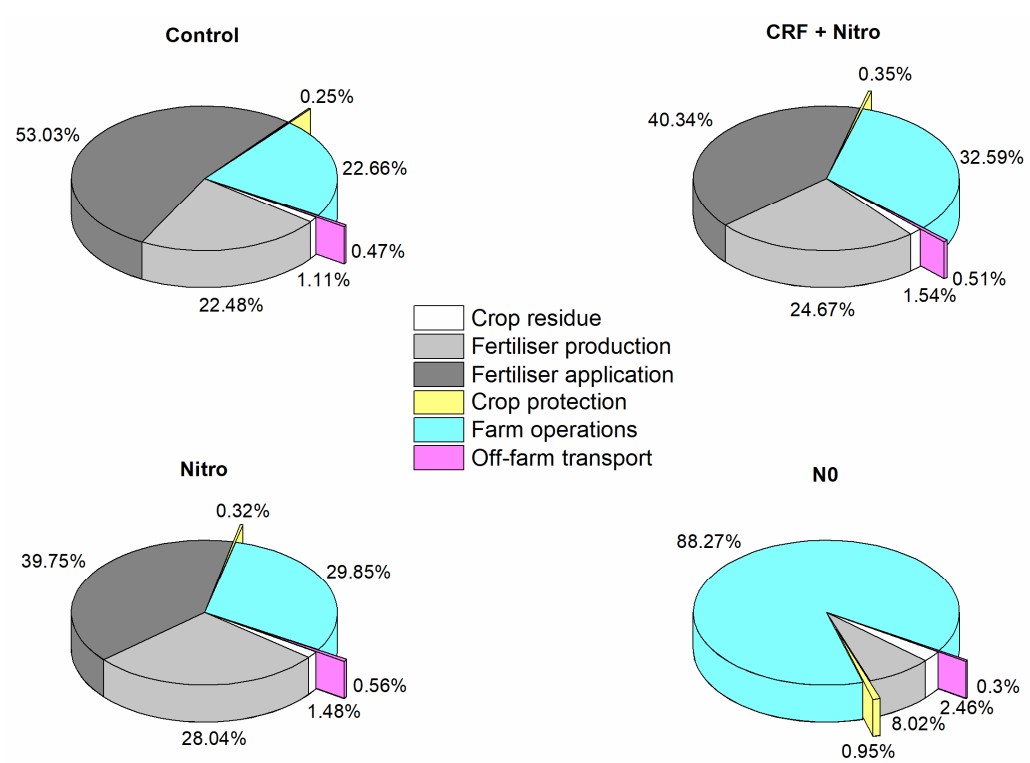

**Figure 6.** Comparison of percentages of GHG emissions from different sources in barley production under different N fertilisation strategies in Larissa.

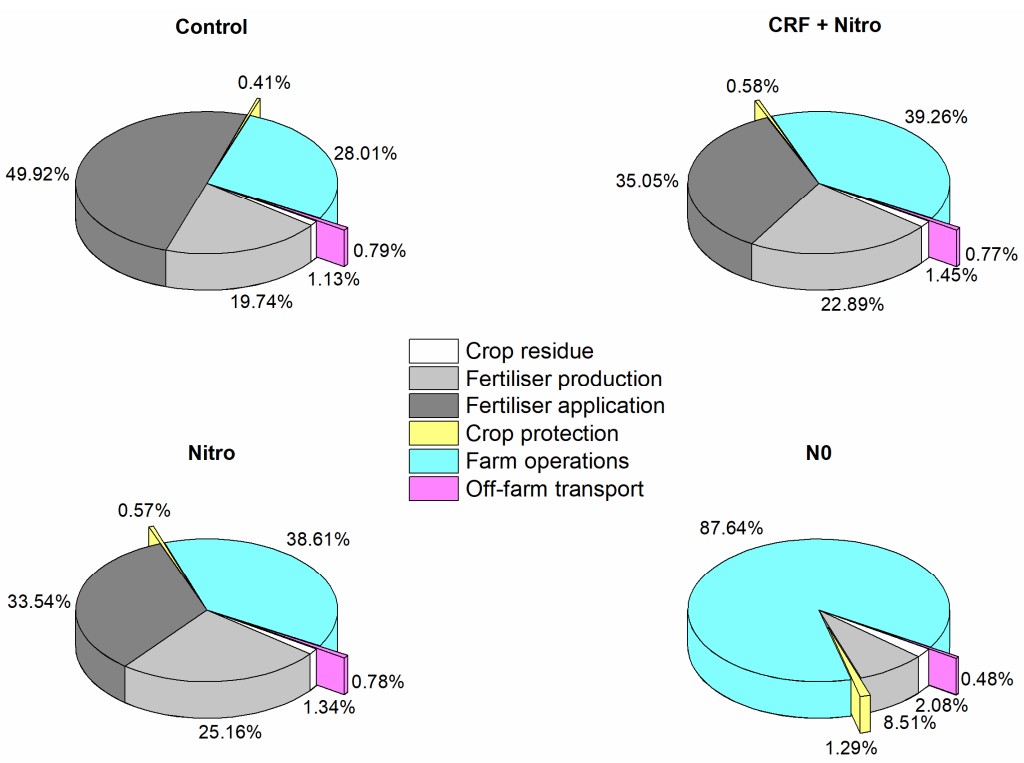

**Figure 7.** Comparison of percentages of GHG emissions from different sources in barley production under different N fertilisation strategies in Livadia.

### 3.5. Economic Analysis

The effectiveness of any crop strategy is ultimately evaluated on the basis of its economics (Figure 8). Net profit was significantly affected by the N fertilisation strategies ($p < 0.05$) and by the interaction location × treatment ($p < 0.001$) and ranged between 448.8 € ha$^{-1}$ and 978.4 € ha$^{-1}$. Statistically significant differences between the low-carbon practices and the control were evident solely in Livadia, exclusively attributed to the Nitro treatment. Compared to control, the net profit increased by 9.0 and 19.8% under the CRF + Nitro and Nitro, respectively, in Livadia. This trend was also evident in Larissa, where the net profit increased by 5.2% under the CRF + Nitro treatment. Notably, the Nitro treatment exhibited a marginal reduction in net profit (i.e., −8%) when compared to the control in this experiment. Differences in total production cost largely depended on fertiliser prices, which contributed 37–39%, 33–34%, and 34–37% of total costs in Control, CRF + Nitro, and Nitro. The control presented the highest production cost, mainly due to the increased fertiliser rate compared to the low-carbon practices. Despite the fact that the total production cost in Livadia was higher compared to Larissa, due to the increased number of tillage operations, higher cost of fertilisers, and fungicide spraying, the treatments in Livadia (the only exception was N0) presented a higher net profit.

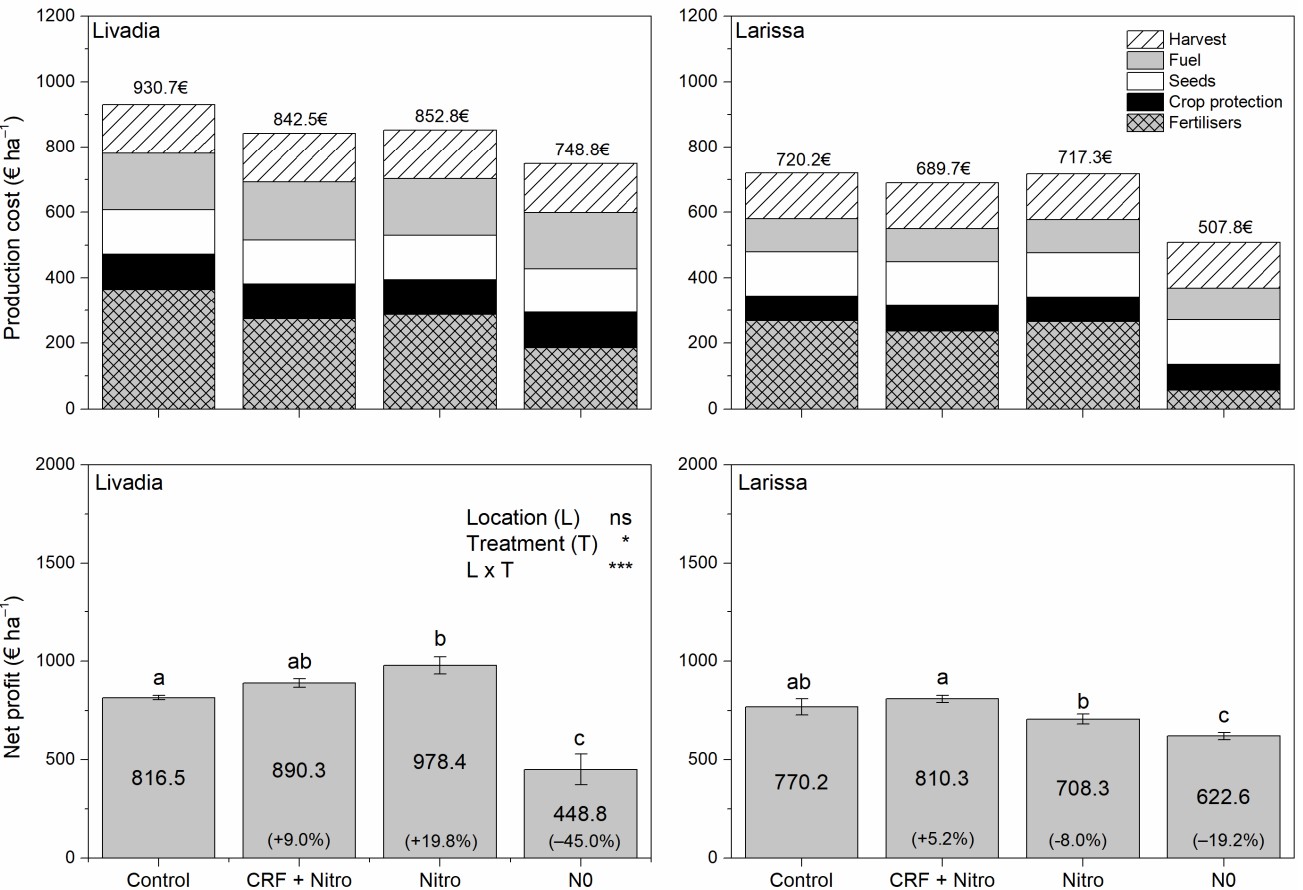

**Figure 8.** Production cost and net profit across the different N fertilisation strategies in Livadia and Larissa. Error bars represent the standard error of the mean ($n = 4$). Different letters indicate statistical significance differences within the same location according to L.S.D. test ($p < 0.05$). Percentage change relative to the control is indicated within parentheses. Analysis of variance (ANOVA) is also shown. * F values significant at the $p < 0.05$ probability levels. *** F values significant at the $p < 0.001$ probability levels.

## 4. Discussion

*4.1. Modulating Barley Growth: The Dynamic Interplay of Nitrogen Dosage, Crop Developmental Stage, and Soil Nitrogen Availability*

It was demonstrated that the impact of nitrogen application at sowing on the extent of barley's green canopy coverage during the tillering stage became evident once it reached an approximate threshold of 18–20 kg N ha$^{-1}$. This threshold aligns closely with earlier research findings demonstrating barley above-ground nitrogen uptake at various developmental stages [72]. The lack of significant differences between the N0 treatment and the control, as well as the other treatments in Livadia, during the tillering stage can possibly be attributed to two key factors. First, the crop in Livadia benefited from an earlier sowing date compared to Larissa (with a 15-day difference), allowing the plants to take advantage of higher temperatures (Figure 1). These elevated temperatures, in turn, promoted greater nitrogen mineralization in the soil [73,74]. Secondly, the higher soil organic content in Livadia, as compared to Larissa, is likely to have played a role in supporting improved nitrogen availability by serving as a potential source of nutrients [75–77]. Interestingly, from the tillering stage onwards, it became apparent that the soil's nitrogen reserves were insufficient to meet the heightened demands of the crop in the N0 treatment. As a result, plants subjected to this treatment exhibited elevated levels of leaf senescence and tiller mortality, a pattern consistent with an earlier study conducted on wheat [78]. This study demonstrated that in nitrogen-stressed treatments, tiller mortality began significantly earlier compared to treatments with no nitrogen limitations.

*4.2. Exploring the Feasibility of a 34% Nitrogen Rate Reduction in Malt Barley without Compromising Yield*

To address this challenge, we followed two distinct strategies with regard to the fertilisers applied. In the first strategy, we formulated a mix consisting of controlled-release fertiliser (CRF) combined with ammonium sulfate nitrate fertiliser (in a 40:60 ratio). This approach was anticipated to deliver a reduction in fertiliser costs, as supported by previous studies [14,42]. It was also expected to enhance nitrogen availability across short (attributable to the nitrate content in the ammonium sulfate nitrate fertiliser), medium (due to both the ammonium sulfate nitrate fertiliser and the CRF), and extended (primarily owing to the CRF) timeframes. In our secondary strategy, we exclusively applied ammonium sulfate nitrate fertiliser. This decision was based on its specifications, which include 7.4% nitrate-N, 18.3% ammonium-N, and 0.3% cyanamide-N. This particular composition enables both rapid and moderate modes of nitrogen release simultaneously, presenting us with a versatile option for managing nitrogen availability compared to common urea (control).

It was demonstrated that under the tested conditions, a 34% nitrogen rate reduction in malted barley did not result in yield penalties. It is important to note that the reduction in N rate was allocated, with approximately 47% assigned to basal fertilisation and about 29% to topdress application at the tillering phase. The rationale behind our decision to distribute the nitrogen reduction in this specific manner between basal and topdress fertilisation is grounded in barley ontogeny. It is extensively documented that across environmental and genotypic sources of variation, grain yield in two-rowed barley, as with other cereals, is more strongly associated with grain number than with grain size [79–82]. Additionally, it is suggested that the critical period for determining grain number coincides with that of stem and spike growth. Hence, it appears that tiller/floret mortality is more critical than tillering and floret initiation [83].

Our strategy regarding the N rate and the types of fertilisers employed facilitated the crop in the low C treatments (CRF + Nitro and Nitro), developing a similar number of spikes m$^{-2}$ and grains per spike compared to the control. Consequently, the yield gap between the control and the tested N strategies was minimized. This was also evidenced by the green canopy cover during the booting stage. Previous research [84] has shown that Canopeo-measured green canopy cover accounted for 65% of the variability in winter

malting barley grain yield. Although the lack of yield penalties remained consistent across both experiments, it appeared that grains spike$^{-1}$ exhibited a higher value in Larissa, whereas spikes m$^{-2}$ recorded a higher value in Livadia. Given the fact that the total N applied in the low-C practices was actually the same in both experiments, the recorded differences can be attributed to the environmental conditions. These may include distinct weather patterns during critical stages of yield component determination, variations in soil types, differences in soil preparation before sowing, and fluctuations in nitrogen rates during basic fertilisation. Additionally, these results may be linked to the commonly reported negative relationship between grains spike$^{-1}$ and spikes m$^{-2}$. According to Serrago et al. [83], when resources that limit crop growth during stem elongation are allocated to the survival of tillers (improving spikes m$^{-2}$), there will be fewer resources available per spike, producing the trade-off in grains spike$^{-1}$ and vice versa.

The reduction in the yield gap can be attributed to the fact that the tested N strategies have led to a substantial increase in nitrogen agronomic efficiency. In fact, the nitrogen agronomic efficiency values in CRF + Nitro and Nitro were mostly higher compared to those reported in the literature [19,72,85,86]. There are two explanations for this performance. Firstly, our results align with the common trend of demonstrating an increase in nitrogen agronomic efficiency through the reduction of total nitrogen applied [20,86–88]. The second reason can be attributed to the interaction Environment × Fertiliser type. Indeed, it was evident that CRF + Nitro and Nitro presented contrasting responses across the two experiments. The inconsistency between the two experiments stems from a difference in the timing of the spring topdress N application. In Livadia, the application was delayed compared to Larissa, primarily due to adverse weather conditions. This delay resulted in a more advanced stage of plant development at the time of application. It is well established that fast-release fertilisers, such as nitrates, provide N in a readily available form that plants can quickly absorb. This becomes crucial during the late season, when barley's demand for nutrients is high and the growth period is limited. In contrast, CRFs, with their unique nutrient release mechanism explained by a sigmoidal curve involving three phases [28], take time to release nutrients. Although this characteristic may not align well with the late-season growth needs of the plant, it proved advantageous when the fertiliser application was conducted at the onset of tillering (GS22) in Larissa.

### 4.3. Decoding the Impact of the Tested Nitrogen Strategies on Grain Protein Content (GPC) and Size

Our findings align with previous evidence [89], demonstrating that the optimal grain nitrogen concentration for achieving maximum grain weight is below the level that maximises grain nitrogen accumulation or protein deposition. Furthermore, it was revealed that under the tested conditions, grain size and grain protein content exhibit contrasting responses in relation to nitrogen rate. A growing body of literature supports the notion that GPC is inversely associated with grain size [25,26,90,91]. According to Bertholdsson [92], mainly two scenarios contribute to GPC dynamics, both involving water stress and nitrogen. In the first scenario, pre-anthesis drought stress results in low nitrogen uptake during the vegetative period, reducing yield potential. This leads to increased nitrogen availability during grain filling due to a lower number of grains, subsequently elevating GPC. In the second scenario, drought stress during late grain filling restricts carbohydrate incorporation in the grain. This means that the total nitrogen content in the grain remains relatively constant, but the grain size decreases. Earlier studies have shown that starch accumulation appears to be more sensitive to post-anthesis stress than nitrogen accumulation [93,94].

Although a 34% reduction in nitrogen rate did not result in yield penalties or a decrease in grain size, it did have a negative impact on GPC, regardless of the type of fertiliser employed. Our previous experiments, conducted across the major malt barley production zones in Greece, demonstrated that maintaining the total nitrogen supply below 100 kg N ha$^{-1}$ led to a decline in GPC under high-yielding conditions [25]. What is also noteworthy is that even the higher nitrogen rates (i.e., Control = 108 kg N ha$^{-1}$) under high

yielding conditions (i.e., Livadia: Control GY = 7.28 t ha$^{-1}$) were insufficient to meet the lower (9.0%) limit for grain protein content, according to the specifications of many brewers worldwide [25]. We hypothesised that, despite the lower nitrogen rate in the CRF + Nitro treatment compared to the control, it could potentially provide more nitrogen during the late season, attributed to the nitrogen from the CRF. Consequently, a positive impact on GPC was anticipated, as indicated by recent research [95,96]. However, the results from both experiments did not confirm this hypothesis. Previous research has shown that the amount of N uptake in the aboveground parts of barley from pre-anthesis accounts for up to 97% of the total grain N at maturity, depending on the variety and environment [97]. Consequently, it appears that GPC is predominantly influenced by the quantity of nitrogen available during the reproductive phase (e.g., from culm elongation to heading). Given that the tested CRF fertiliser had a 2–3 month release period, it is highly likely that a particular amount of nitrogen was released outside the aforementioned developmental window. Therefore, we believe that a product with a faster release period (e.g., 1–2 months) would perform better under the tested conditions regarding its impact on GPC.

Interestingly, the reduction in nitrogen rate had a negligible effect on the retention fraction for both the CRF + Nitro and Nitro treatments, as well as the N0 treatment. This can be attributed to the favourable meteorological conditions—characterised by adequate precipitation (>60 mm) and moderate temperatures—that prevailed during the period when grain size was determined (Figure 1). As recently demonstrated [52], cumulative rainfall from 6 days pre-anthesis to 20 days post-anthesis emerges as a dominant factor (or the most limiting factor) in determining grain size in malted barley grown in Mediterranean environments. Consequently, it has the capacity to offset the impact of other stresses [26,98]. In the case of the N0 treatment, besides the aforementioned explanation, the absence of nitrogen through fertilisation resulted in fewer grains m$^{-2}$. It is well established that between grains m$^{-2}$ and grain size in cereals there is a clear negative relationship [25,82,99–101].

### 4.4. The Environmental Impacts

The total GHG and carbon footprint values in this study fall within the lower range compared to those reported for barley in the literature [9,14,102–105]. According to Niero et al. [103], making a direct comparison among the results of various life cycle assessment (LCA) studies is not always straightforward due to differences in system boundary definitions and assumptions. Nevertheless, the primary focus of the current research was to investigate the variation in carbon footprint and GHG emissions across the tested N strategies.

Our findings align closely with a growing body of literature, highlighting a consistent trend wherein synthetic nitrogen fertilisers, employed in barley crop production, emerge as the primary contributors to total GHG emissions [8–12,14]. Under the tested conditions, the contribution of nitrogen fertilisers (i.e., emissions from the production and application of N) to total GHG emissions exceeded 75% regarding the conventional fertilisation practice employed by farmers (control) in Larissa. Therefore, removing nitrogen fertilisation from the crop system could provide insight into the maximum theoretical reduction of total GHG emissions in malt barley production. These theoretical limits were found to range from 68.5% to 74.3% in the tested conditions. However, implementing a 34% reduction in nitrogen dosage, as seen in the CRF + Nitro and Nitro treatments, resulted in a reduction of GHG emissions by approximately 30% and 26% in the CRF + Nitro and Nitro treatments, respectively. This implies that the mitigation of GHG emissions does not exhibit a linear relationship with the reduction in nitrogen rate.

Mixing CRF with ammonium sulfate nitrate (CRF + Nitro) emerged as the most efficient strategy for mitigating greenhouse gas (GHG) emissions in malt barley production. In comparison to the control (common sulfur urea at 40% N), CRF + Nitro reduced GHG emissions from nitrogen fertiliser application by approximately 45%. A recent study in wheat [14] demonstrated that, during on-field nitrogen fertiliser application, a blend of CRF with common urea led to a reduction in N$_2$O emissions, resulting in 16–35% lower GHG

emissions than common urea alone. A meta-analysis conducted by Akiyama et al. [32], based on 113 datasets from 35 studies, reported that the application of CRF reduced $N_2O$ emissions by 35% compared to conventionally used nitrogen fertiliser. Furthermore, a global meta-analysis based on 866 observations of 120 studies indicated that application of CRF instead of urea (same N rate) significantly decreased nitrous oxide ($N_2O$) emissions, N leaching, and ammonia ($NH_3$) volatilization by 23.8%, 27.1%, and 39.4%, respectively [106]. From another perspective, it has also been tested the effect of synthetic N fertiliser replacement by cow manure. The results showed that, compared with conventional nitrogen application, replacement of synthetic N by 20 and 50% with cow manure reduced $N_2O$ emissions by 6.65 and 11.65%, respectively [107].

The superior performance of CRF + Nitro over Nitro can be attributed to lower GHG emissions occurring both during the production and application stages of the fertilisers. As demonstrated in previous studies [7,9,61], an increase in greenhouse gas (GHG) emissions due to the implementation of a specific management practice does not necessarily equate to a larger carbon footprint. This depends on whether the cropping practices result in greater grain yields. Such a scenario was observed in the Nitro treatment in Livadia. Despite Nitro yielding higher GHG emissions compared to CRF + Nitro, its carbon footprint was lower because, under specific conditions (e.g., soil and meteorological conditions, timing of fertiliser application in relation to barley developmental stage, etc.), Nitro (ammonium sulfate nitrate) proved to be a more effective fertiliser for stimulating GY (for more details, see Section 4.2).

### 4.5. The Economic Benefits

Economic profit serves as the primary motivator for farmers when selecting a management practice. Our study confirms previous evidence suggesting that the economic performance of CRFs could be optimised by blending them with common fertilisers [13,14,42]. As shown in Figure 8, CRF + Nitro not only had the lower production cost but also presented the highest net profit among the tested treatments. The only exception was in Livadia, where Nitro recorded the highest net profit, yet there was no statistically significant difference between CRF + Nitro. Compared to the control (common sulphur urea), the cost of CRF was 14% higher, whereas the cost of Nitro (ammonium sulfate nitrate) was 4.7% less.

Interestingly, there is an expectation that the price of CRFs will continue to decrease in the future. Currently, CRF production is often carried out in small batches; however, this cost is likely to be mitigated as CRFs become more abundant in the global market and production volumes increase [28]. As a result, it is anticipated that CRF + Nitro will yield greater net profit for the producers compared to common sulphur urea.

### 4.6. Limitations of the Study

Although combining CRF-containing urea with ammonium sulfate nitrate has shown significant potential for enhancing yield and improving environmental and financial performance, its usage still faces two critical limitations. Firstly, urea and ammonium nitrate (or ammonium sulfate nitrate) are deemed incompatible for solid blend preparation. This is attributed to the highly hygroscopic double salts they form upon contact, causing the mixture to quickly become wet and absorb moisture, ultimately resulting in the formation of a liquid or slurry [108]. Nevertheless, there are strategies to address this limitation. On one hand, the fertiliser industry has developed techniques to prepare safe blends of urea and ammonium nitrate [109]. On the other hand, the industry has been investigating whether the coating materials of CRFs offer protection against the aforementioned incompatibility (personal communication).

The second constraint producing a blend of CRF containing urea with ammonium sulfate nitrate could be due to the size incompatibility of the raw materials, potentially leading to an uneven distribution of nutrients on plants. While not a significant challenge for the fertiliser industry to overcome, the application of CRF + Nitro in the experimental

sites required two separate passes—one for the CRF containing urea and another for the ammonium sulfate nitrate—to ensure even spreading.

## 5. Conclusions

It was shown that, although a 34% reduction in the N rate did not result in yield penalties or a decrease in grain size, it did have a negative impact on GPC. In fact, our findings confirm previous evidence, suggesting that the optimal grain nitrogen concentration for achieving maximum grain weight is below the level that maximises grain nitrogen accumulation or protein deposition. While the current trend for enhancing CRF performance involves creating blends with common urea, our study revealed that further optimisation is possible by replacing common urea with a nitrate-containing fertiliser. Indeed, CRF + Nitro not only reduced CF by approximately 30% compared to the control but also increased N agronomic efficiency by 51.5% and net profit by 7.1%. Our study shed light on the effects of the interaction Environment × Fertiliser type × Time of spring topdress N application on all sustainability aspects (i.e., GY, GPC, grain size, environmental impact, and economic viability). Furthermore, it was demonstrated that the maximum achievable reduction in total GHG emissions and CF, by excluding N fertilisation from the crop system, ranged from 68.5% to 74.3% for GHG emissions and 53.8% to 67.1% for CF. However, implementing a 34% reduction in nitrogen dosage, as seen in the CRF + Nitro and Nitro treatments, resulted in a reduction of GHG emissions by approximately 30% and 26% in the CRF + Nitro and Nitro treatments, respectively. This implies that the mitigation of GHG emissions does not exhibit a linear relationship with the reduction in nitrogen rate.

**Supplementary Materials:** The following supporting information can be downloaded at: https://www.mdpi.com/article/10.3390/agriculture13122272/s1, Table S1: Chemical and physical soil properties at the beginning of the experiments in Livadia and Larissa (2021); Table S2: Field plot operations throughout the malt barley growing season in Livadia and Larissa.

**Author Contributions:** Conceptualization, P.V.; methodology, P.V.; field, laboratory work; P.V., A.S. and V.K.; resources, V.K.; data curation, P.V.; writing—original draft preparation, P.V.; writing—review and editing, P.V., A.S. and V.K.; visualisation, P.V.; funding acquisition, V.K. All authors have read and agreed to the published version of the manuscript.

**Funding:** This research was funded by the Athenian Brewery S.A. (Heineken Group).

**Institutional Review Board Statement:** Not applicable.

**Data Availability Statement:** The data presented in this study are available upon request from the corresponding author.

**Conflicts of Interest:** Author V.K. was employed by the company Athenian Brewery S.A. The remaining authors declare that the research was conducted in the absence of any commercial or financial relationships that could be construed as a potential conflict of interest. The authors declare that this study received funding from Athenian Brewery S.A. The funder was not involved in the study design, collection, analysis, interpretation of data, the writing of this article, or the decision to submit it for publication.

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
