# Peer review of "Optimizing Sustainability in Malting Barley: A Practical Approach to Nitrogen Management for Enhanced Environmental, Agronomic, and Economic Benefits"

_agriculture, doi:10.3390/agriculture13122272_

Round 1
Reviewer 1 Report
Comments and Suggestions for Authors
The manuscript “Optimizing Sustainability in Malting Barley: A practical Approach to Nitrogen Management for Enhanced Environmental, Agronomic, and Economic benefits” involves the impacts of fertiliser source on Yield, Environmental, Agronomic, and Economic benefits. The research has good originality, experimental design is appropriate, and conclusions supported by the results.
several information should be revised and improved before publication.
Materials and method
L98, Please provide the GHG calculation process
L131 What is the release period of controlled release fertilizer
L172 NAE = YN – Y0 / AN, should be NAE =(YN – Y0)/AN,
L187, Carbon Footprint (CF) Calculation equation should be show
Results and Discussion
L294-296,the application of CRF + Nitro have the highest NAE in Livadia. while, Nitro treatment exhibited the highest NAE in Livadia., Provide reasons should be in the discussion section.
L301 table 2, CRF + Nitro and Nitro treatment have the same N rate, but grains spike-1 displaying statistically significant differences in Livadia and Larissa, reasons should be explained in the Discussions and Conclusions
Reviewer 2 Report
Comments and Suggestions for Authors
Agriculture-2717776 Comments
The MS is interested in the management changes in the N source that cause the CF to decrease. However, the MS has some questions below;
1.Why did the author choose the 34% reduction N rate for the experiment?
2. Please include a discussion of the relationships between GHG and CF, as well as those N treatments and "other types of N."
3. Incorporate the Net Profit relating to GHG and CF, along with yield management, into the Discussion and Conclusion topic.
Reviewer 3 Report
Comments and Suggestions for Authors
Figure 4. Arrange the letters for statistically significant differences correctly (from smallest value to largest) for Larissa. It is: Control – a, CRF + Nitro – b, Nitro – c and should be Control – a, CRF + Nitro – c, Nitro – b.
Please very much explain whether the barley grain received met the quality requirements for brewing purposes.
In my country, the maximum dose of nitrogen in the cultivation of malting barley is about 75 kg/ha.
Reviewer 4 Report
Comments and Suggestions for Authors
IT is a result of a just one growing season, with no reduction on yield of barley. AND, how about the second growing season ? It should be discuss or mentioned in the manuscript. As we see, we have found many studies showing no yield reduction for one growing season with N fertilizer reduction.
